# Role of Hepatic Aryl Hydrocarbon Receptor in Non-Alcoholic Fatty Liver Disease

**Nikhil Y. Patil** [1][ID]**, Jacob E. Friedman** [2][ID] **and Aditya D. Joshi** [1,2,*]

1    Department of Pharmaceutical Sciences, University of Oklahoma Health Sciences Center, Oklahoma City, OK 73117, USA

2    Harold Hamm Diabetes Center, University of Oklahoma Health Sciences Center, Oklahoma City, OK 73117, USA

*    Correspondence: aditya-joshi@ouhsc.edu

**Abstract:** Numerous nuclear receptors including farnesoid X receptor, liver X receptor, peroxisome proliferator-activated receptors, pregnane X receptor, hepatic nuclear factors have been extensively studied within the context of non-alcoholic fatty liver disease (NAFLD). Following the first description of the Aryl hydrocarbon Receptor (AhR) in the 1970s and decades of research which unveiled its role in toxicity and pathophysiological processes, the functional significance of AhR in NAFLD has not been completely decoded. Recently, multiple research groups have utilized a plethora of in vitro and in vivo models that mimic NAFLD pathology to investigate the functional significance of AhR in fatty liver disease. This review provides a comprehensive account of studies describing both the beneficial and possible detrimental role of AhR in NAFLD. A plausible reconciliation for the paradox indicating AhR as a 'double-edged sword' in NAFLD is discussed. Finally, understanding AhR ligands and their signaling in NAFLD will facilitate us to probe AhR as a potential drug target to design innovative therapeutics against NAFLD in the near future.

**Keywords:** AhR; aryl hydrocarbon receptor; CD36; cluster of differentiation 36; Cyp1a1; cytochrome p450 1a1; NAFLD; non-alcoholic fatty liver disease; NASH; non-alcoholic steatohepatitis; XRE; xenobiotic response element; TCDD; 2,3,7,8-tetrachlorodibenzo-p-dioxin

## 1. Introduction

The Aryl hydrocarbon Receptor (AhR) is a ubiquitously expressed, ligand activated transcription factor known to play a diverse role in physiological and plethora of toxico-pathological conditions. This includes cell proliferation and death [1,2], developmental biology [3], immunology [4], nuclear hormone signaling [5], dioxin toxicity [6], carcinogenesis [7], cardiotoxicity [8] and hepatotoxicity [9]. Recently, a significant number of studies investigating the role of AhR in metabolic diseases, including non-alcoholic fatty liver disease (NAFLD) have been conducted. NAFLD affects more than 25% of the US and global population, and is associated with significant morbidity and mortality due to complications of liver cirrhosis, hepatic decompensation and hepatocellular carcinoma [10]. NAFLD, a chronic liver disease, is characterized by excessive hepatic fat accumulation and insulin resistance without uptake of alcohol [11,12]. Despite multiple previous reviews interrogating the role of nuclear receptors in NAFLD, a comprehensive report illustrating involvement and effect of AhR signaling in NAFLD pathophysiology is absent [13,14]. Moreover, the role of AhR in fatty liver disease has been controversial due to both beneficial and adverse effects on liver pathology [15,16]. Therefore, the focus of this review is to depict our current understanding of AhR signaling in NAFLD.

## 2. Aryl Hydrocarbon Receptor

Initial work in AhR biology began in 1976, when Alan Poland and colleagues identified induction of an Aryl hydrocarbon hydroxylase enzyme activity (now known as cytochrome

P450 enzyme 1a1, Cyp1a1) by polycyclic and halogenated aromatic hydrocarbons (PAHs and HAHs) [17–21]. It was also hypothesized that the Cyp1a1 induction was mediated by an inducible proteinic receptor [21–23]. This receptor identified and later named as AhR was successfully cloned in both mice and humans in early 1990s [23–29]. Historical studies on the AhR focused their efforts towards understanding the molecular basis for the adaptive and toxic responses to variety of chemical pollutants including prototypical 2,3,7,8-tetrachlorodibenzo-p-dioxin (TCDD) and benzo[a]pyrene, which manifests as a broad spectrum of biological processes including immune-, cardio- and hepatotoxicity, wasting syndrome, liver and soft tissue tumor, and endocrine disorders [2,5,6,30–39]. For the detailed historical perspective regarding discovery of AhR and early toxicological studies, we refer readers to the reviews by Nebert DW [40] and Jackson DP, Joshi AD and Elferink CJ [34].

Multiple studies have portrayed AhR's canonical signaling pathway [5,31]. In the absence of a ligand, AhR resides in the cytoplasm in association with chaperonins—p23, AhR interacting protein, and heat shock protein 90 [33,34,36]. Upon binding with an archetypical agonist such as TCDD, AhR translocate to nucleus, dissociates from the chaperonins, and heterodimerizes with Aryl hydrocarbon Receptor nuclear translocator (Arnt) [41]. The agonist-bound AhR-Arnt dimerized complex interacts with the promoter region of AhR responsive genes. Classic AhR targets encompass genes involved in phase I metabolism, including Cyp1a1 and phase II enzymes (quinone oxidoreductase, aldehyde dehydrogenase, glutathione-s-transferase, etc.) [36,42,43].

AhR belongs to the class I basic helix-loop-helix Per-Arnt-Sim (bHLH/PAS) family of transcription factors and contains a basic, helix-loop-helix, PAS A and PAS B domains critical for molecular functions [5,31,44–46]. The basic region is essential for DNA binding, whereas the HLH and PAS A domains are vital for protein–protein interactions. Conformational changes in the PAS A domain facilitate nuclear translocation of AhR. Various biochemical and biophysical studies have confirmed that the PAS B is the de facto ligand-binding domain [34,44,46]. Although structural determination of AhR has made significant progress including availability of a partial crystal structures containing bHLH and PAS A domains of AhR-Arnt heterodimer in complex with XRE, Drosophila AhR PAS B domain, and recently published Cryo-EM structure of the indirubin bound AhR-Hsp90-XAP2 complex in cytoplasm – a complete three-dimensional crystal structure of AhR has not yet been solved [47–49].

Apart from exogenous ligands of anthropic origin including members of halogenated and polycyclic aromatic hydrocarbons including TCDD and Benzo[a]pyrene, exogenous AhR ligands of natural origins from plants and vegetables such as flavonoids, resveratrol, luteolin, genistein have also been recognized [50]. Moreover, endogenous AhR agonists generated from the host metabolism (kynurenine, cinnabarinic acid, tryptamine, 6-formylindolo(3,2-b)carbazole, bilirubin) and detected in their commensal microflora (indole-3-carbinole, indole-3-acetic acid, indole[3,2-b]carbazole 7-ketocholesterol) have been identified and extensively studied to understand pathophysiological role of AhR signaling in various metabolic disorders [50–52]. This includes how the AhR acts as a sensor for endogenous tryptophan metabolites generated from the microbiome [51]. For an encyclopedic review of the structurally and functionally diverse AhR ligands, we refer the readers to prior reviews by Denison and Nagy [31] as well as by Nguyen and Bradfield [50].

## 3. Generation of Pioneering AhR Knockout Models

Upon cloning and characterization of the mouse AhR sequence, AhR knockout mouse lines (referred as: AhR KO, AhR null, AhR$^{-/-}$) were independently constructed by various research groups [53–55]. As expected, AhR KO mice showed resistance to TCDD and Benzo[a]pyrene toxicity [55,56]. Interestingly, AhR null mice generated by the deletion of exon 1 exhibited biliary inflammation and fibrosis around portal triad when fed with a normal diet [53]. However, deletion of exon 2 resulted in the milder liver pathology exhibiting cholangitis and mild fibrosis around ducts [54,57]. Although the molecular basis

for the phenotypical differences observed between the AhR null strains are not clearly understood, the differences are attributed to the genetic background and/or the gene targeting strategies affecting other genes [57]. Nevertheless, this was the first in vivo observation that the deficiency of AhR results in hepatic fibrosis and a host of other pathological conditions including failure of developmental closure of the ductus venosis [58], cardiac hypertrophy and fibrosis [59–62], impaired fertility, stunted postnatal growth, multi-organ dysregulation of organogenesis during in utero development, higher risk of embryonic death [63–66], altered mammary gland development [67,68], decreased barrier function [69], immune dysfunction [70–72], oculomotor deficiencies and neuronal function disorders [73,74]. Overall, results suggested a critical role of AhR in developmental as well as pathophysiological processes.

## 4. AhR and Liver Fibrosis Models

Apart from hepatocytes, a diversity of cell type plays a significant role in hepatic diseases, including infiltrating innate and adaptive immune cells, endothelial cells, stellate cells, and Kupffer cells. In human liver, the mRNA expression of AhR target genes CYP1A1 and CYP1A2 is decreased in patients with hepatic fibrosis [75]. In a recent study, Yan et al. observed that AhR was expressed at a higher level in quiescent hepatic stellate cells than in the activated stellate cells [76]. Moreover, activation of AhR with an endogenous agonist— 2-(1′H-indole-3′-carbonyl) thiazole-4-carboxylic acid methyl ester (ITE) was able to provide protection against $CCl_4$-induced fibrosis [76]. These studies supported the observations that the downregulation of AhR signaling promotes fibrosis, whereas activation of AhR by endogenous agonist confer protection. On the contrary, Hoshi et al. showed that $CCl_4$ treatment leads to elevation of endogenous AhR agonist kynurenine, which activates AhR signaling and promotes fibrosis [77]. Moreover, TCDD treatment for 2 weeks elevated the hepatic expression of fibrotic markers and a 6-week TCDD regimen induced liver fibrosis in mice in an AhR-dependent manner [78]. In another study, Il22ra1 knockout mice exhibited reduced fibrosis in response to thioacetamide and $CCl_4$. Blocking Il22 or Il17 production using the AhR antagonist, CH223191 resulted in reduced fibrosis [79]. Therefore, a role of AhR signaling in hepatic fibrogenesis is critical but complex, and additional studies are required to completely uncover AhR function in liver fibrosis.

## 5. AhR Signaling Promotes Hepatic Steatosis and NAFLD Pathology

To study the effect of AhR activation in vivo, a tetracycline-inducible constitutively active AhR (CA-AhR) mouse was constructed in the Xie laboratory [15]. Activation of AhR showed decreased body mass and resulted in the induction of spontaneous steatosis characterized by an accumulation of liver triglycerides but not cholesterol. Further microarray analysis indicated that the expression of fatty acid translocase protein, Cd36 (cluster of differentiation 36) was elevated in CA-AhR transgenic mice. CD36 is known to facilitate transport of long-chain fatty acids and is regulated by pregnane X receptor, liver X receptor and proliferator-activated receptor γ [80]. Using electrophoretic mobility shift and luciferase activity assays, Lee et al. confirmed Cd36 as a novel transcriptional target of AhR, and showed inhibition of hypertriglycedemia in response to TCDD treatment due to attenuation of free fatty acid uptake in Cd36 knock-out mice [15]. Constitutive activation of AhR (CA-AhR) inhibited mitochondrial β-oxidation, increased adipose triglyceride lipase, decreased white adipose tissue fat mass, and increased hepatic oxidative stress [15]. In another study, constitutively activated human AhR transgenic mice subjected to high-fat diet containing 60 kcal% fat for 12 weeks displayed exacerbated steatosis [81]. Both hepatic triglyceride and cholesterol content were significantly higher in the transgenic mouse model containing constitutively active human AhR. Despite increased steatohepatitis, the transgenic mice were protected from high-fat diet induced obesity and showed improved insulin sensitivity. This study identified hepatokine Fgf21 as a direct AhR target and indicated that in the constitutively active AhR mice, circulating concentration and hepatic expression of FGF21 was upregulated. Whereas knocking down Fgf21 using adenoviral expression

of short hairpin RNA targeting FGF21 in high-fat diet fed transgenic mice resulted in the mitigation of steatosis but exacerbation of hepatic injury and inflammation [81].

Apart from the use of constitutively active AhR mouse models, various groups have exploited AhR knockout systems to understand the role of AhR signaling in NAFLD. The Tischkau laboratory showed improved insulin sensitivity and glucose tolerance in AhR KO mice on a chow diet [82]. Moreover, AhR deficiency protected against a high-fat diet induced steatosis, obesity and inflammation. The hepatoprotective effects resulted from the downregulation of Cd36 and inhibition of lipid synthesis in AhR KO mice [83]. A novel tamoxifen-inducible liver-specific AhR conditional knockout mouse model (AhR-iCKO) was constructed to analyze specifically the effects of hepatocyte-targeted AhR loss in adult mice by avoiding complexities involved due to the loss of AhR during embryonic development [84]. The data from the high-fat diet fed control and inducible AhR knockout mice suggested that the inducible female knockout mice were resistant to weight gain and hepatic steatosis, whereas males were not protected from hepatotoxicity – clearly indicating a sexual dimorphism. The loss of AhR resulted in increased hepatic FGF21, offering hepatoprotection and increased energy expenditure [84]. Similar to the anti-steatotic properties exhibited by the whole body and inducible hepatocyte-specific AhR conditional knockout mice, knockout of the AhR in preadipocytes protected mice from high-fat diet induced obesity and liver steatosis suggesting a role for AhR in adipose tissue and possible cross-talk with liver [85].

The aforementioned studies involving genetic manipulation of AhR thus indicated that the activation of AhR will aggravate the NAFLD pathology, whereas inhibition of AhR signaling using AhR antagonists will alleviate NAFLD in pre-clinical models. Accordingly, high-fat diet fed WT mice chronically exposed to TCDD showed a significant increase in hepatic triglyceride content due to stearoyl coenzyme decarboxylase 1 (Scd1) upregulation and elevated de novo lipogenesis [86]. AhR activation with polycyclic aromatic hydrocarbon and a prototypical AhR agonist, benzo[a]pyrene resulted in an induction of Cyp1a1, which rapidly metabolized estrogen receptor ligand, 17β-estradiol and inhibited the protective effects of estrogen signaling, leading to the NAFLD pathology including hepatic steatosis characterized by triglyceride accumulation and hepatotoxicity in a high-fat diet model [87,88]. Moreover, an endogenous AhR agonist derived from a tryptophan catabolism pathway – kynurenine induced hepatic Cyp1b1 and Scd1 expression and resulted in hepatosteatosis [89], whereas inhibition of AhR activity with α-naphthoflavone showed attenuation of steatosis in both high-fat diet fed in vivo and oleic acid-treated HepG2 models of NAFLD. α-naphthoflavone treatment reduced oxidative stress and insulin resistance as well as mitigated NAFLD by modulation of AhR regulated Cyp1a1 and TNFα pathways [90]. A recent study by the Tomlinson's group showed prevention of weight gain in mice that were on a 40-week high-fat diet and α-naphthoflavone regimen. Inhibition of AhR by α-naphthoflavone downregulated expression of Cyp1b1, Scd1, Spp1 and Pparα target genes which otherwise were significantly upregulated in high-fat diet only cohort [91]. Similar to α-naphthoflavone, the use of another AhR antagonist, CH223191 significantly reduced obesity and ameliorated hepatic steatosis in Western diet fed WT mice [92].

## 6. However, AhR Signaling Also Attenuates NAFLD

In opposition to the notion that activated AhR exacerbates hallmarks of NAFLD and genetic or pharmacological inhibition of AhR protects against fatty liver disease, recent studies have indicated the hepatoprotective role of induced AhR signaling in NAFLD models. Krishnan et al. showed that gut-microbiota derived tryptophan metabolites, tryptamine and indole-3-acetate mitigated fatty acid stimulated production of pro-inflammatory cytokines in macrophages [93]. In hepatocytes, indole-3-acetate alleviated lipogenesis by downregulating expression of fatty acid synthase and sterol regulatory element-binding protein 1c (Srebp1c) in an AhR dependent manner [93]. Administration of indole-3-acetic acid, a gut-microbiota derived metabolite from tryptophan and a well-known AhR agonist

obliterated NAFLD parameters by attenuating hepatic lipogenesis, oxidative and inflammatory stress. Indole-3-acetic acid treatment downregulated expression of Srebp1, Scd1, Pparγ, acetyl-CoA carboxylase 1 (Acaca), and glycerol-3-phosphate acyltransferase, mitochondrial (Gpam) as well as mitigated reactive oxygen species (ROS), malondialdehyde levels (MDA), superoxide dismutase activity (SOD) and glutathione (GSH) content in the livers of high-fat diet fed WT mice [94]. A noteworthy study by Xu et al. showed that sulforaphane alleviates hepatic steatosis in mice. Sulforaphane elevated serum and liver levels of indole-3-acetic acid by modulating gut microbiota [95]. Sulforaphane, thus directly or indirectly activated AhR and protected against palmitic-acid induced in vitro model of NAFLD by downregulation of Srebp1c pathway [95]. Indole has shown to alleviate diet induced hepatic steatosis, and the hepatoprotection was dependent on the activation of the AhR signaling pathway [96]. Our laboratory has recently published that a tryptophan catabolite and an endogenous AhR agonist, cinnabarinic acid (CA) protected against both oleic/palmitic acid treated in vitro and high-fat diet induced in vivo models of NAFLD. CA treatment significantly lowered body mass gain and decreased hepatic steatosis both before and after the established NAFLD. CA decreased free fatty acid uptake by downregulation of Cd36 expression as well as attenuated lipogenesis [97]. CA- induced AhR was unable to interact with the Cyp1a1 promoter and therefore did not increase its expression in isolated primary hepatocytes or in vivo [1,98]. However, CA upregulated a novel AhR target gene, stanniocalcin 2 (Stc2) [1,98–101]. Knocking-down either AhR or Stc2 failed to exert hepatoprotective effects by CA in vitro, indicating that CA-mediated protection was dependent on AhR-STC2 signaling pathway [97]. In a methionine-choline-deficient (MCD) mice model, 3, 3'-diindolylmethane (DIM) treatment protected against hepatic steatosis and inflammation as well as shifted the Th17/Treg imbalance to Treg dominance. Protective effects of DIM subsided when AhR was blocked with AhR antagonist CH223191, indicating the role of AhR in DIM mediated protection against NAFLD [102]. An elegant study utilized hepatocyte-specific AhR knockout mouse model (AhR-hKO) and showed that the absence of AhR in hepatocytes accelerated high-fat diet induced hepatic steatosis, inflammation, and injury. This study further identified suppressor of cytokine signaling 3 (Socs3) as a direct transcriptional target of AhR and confirmed that AhR plays a protective role against high-fat diet induced-lipotoxicity via regulation of Socs3 [16].

### 7. The Yin–Yang of AhR Protection against NAFLD—A Conundrum!

Thus far, studies suggest that AhR is a 'double-edged sword' within the context of the fatty liver disease (Table 1). Alterations in AhR expression by genetic or pharmacological approaches in various in vitro and in vivo NAFLD models have indicated that AhR activation is both beneficial or detrimental to the NAFLD pathology (Figure 1) based on various cellular, molecular, biochemical, and epigenetic factors including: (1) Structure and specificity of AhR ligands – which play a critical role in AhR binding to specific gene promoters and subsequent activation of signaling pathways [103]. These ligands are selective AhR modulators that exhibit tissue and cell specific AhR agonist and antagonist activities leading to favorable or unfavorable outcome in NAFLD [104,105]. (2) Upon binding to diverse ligands, AhR undergoes conformational change and interacts with various tissue/cell specific cofactors [34,106]. Apart from canonical AhR-Arnt interaction at XRE, Arnt-independent interaction of AhR with several cofactors including KLF6, CPS1 at non-canonical XRE (NC-XRE) motifs present in the promoter region of p21, Pai1, and Padi2 genes have been identified [34,107–110]. Similarly, direct interaction of AhR and RelA at the novel AhR/RelA response elements present on the promoter regions of NF-kB target genes results in the activation of c-myc and Il6 [111,112]. AhR is also known to form a co-repressor complex with pRb and E2F and suppress expression of S phase genes [113–115]. Therefore, complex interactions of AhR with coactivators and corepressors which regulate multiple signaling pathways, likely have a major influence on protective versus detrimental role of AhR in NAFLD pathology. (3) The ligand and cofactor binding also impact specific post-translational modifications at AhR-bound chromatin [116–118]. We have identified

endogenous AhR agonist, CA specific binding of chromatin modification 'writers' activating transcription factor 2 (Atf2), disruptor of telomeric silencing 1-like histone lysine methyltransferase (Dot1l) and 'reader' metastasis associated protein 2 (MTA2) to AhR at Stc2 promoter [101,119]. Cross-linking chromatin immunoprecipitation coupled mass spectrometry analysis detected CA-specific Atf2 driven histone H4 K5acetylation and Dot1l mediated H3 K79methylation exclusively at the Stc2 promoter [101]. These epigenetic modifications, which were observed in response to CA but not upon TCDD treatment, have been known to decrease DNA-histone interactions, open the chromatin structure and lead to the AhR-mediated transcription activation of Stc2 without Cyp1a1 induction [101]. The CA-induced AhR-mediated Stc2 induction is thus protective against steatosis, inflammation and liver injury observed in NAFLD [97]. (4) AhR signaling is also known to cross react with other signaling pathways involved in lipogenesis and oxidative metabolism. AhR modulates estrogen receptor signaling directly by AhR-Arnt dimer suppressing estrogen receptor-mediated gene expression or indirectly by steric hindrance due to binding of AhR-Arnt complex close to the estrogen receptor elements [120–122]. Similarly, XRE binding by AhR-Arnt complex is in close proximity to the antioxidant response elements (ARE) present in the promoter region of phase II metabolites including NAD(P)H quinone dehydrogenase 1, glutathione-s-transferase, UDP-glucuronosyltransferases—regulated by Nrf2/Maf heterodimer [123–127]. Both estrogen and Nrf2 pathways have been implicated to play a critical role in NAFLD, interact with AhR signaling, and thus affect AhR's function in NAFLD pathogenesis [128,129].

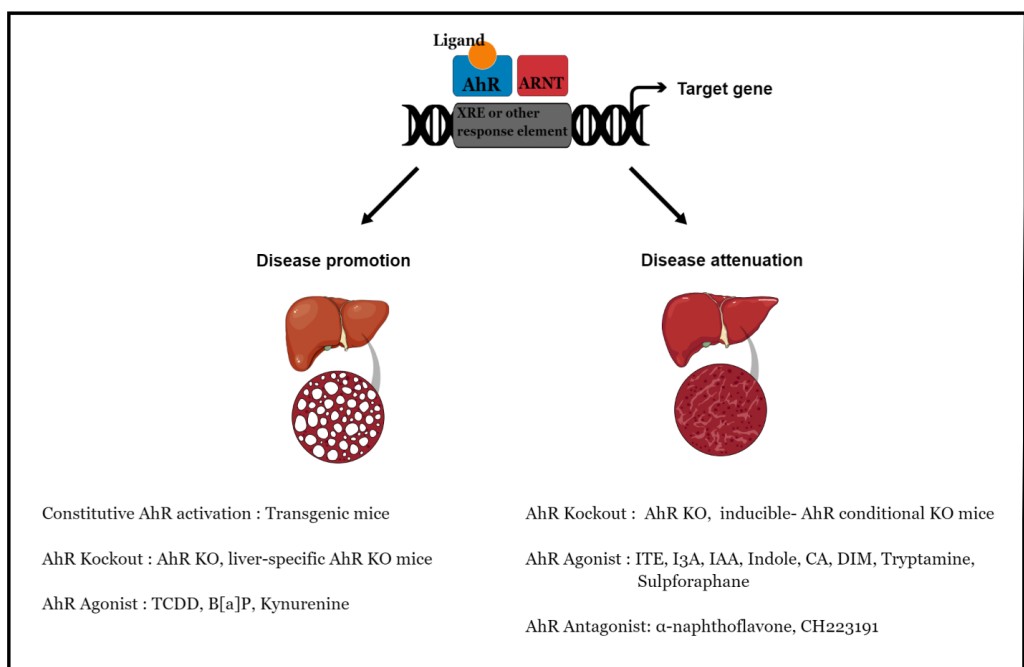

**Figure 1.** Dual role of AhR in promotion and/or attenuation of non-alcoholic fatty liver disease. Alterations in AhR expression by genetic and pharmacological approaches in various in vitro and in vivo NAFLD models have shown hepatotoxic as well as hepatoprotective role of AhR against NAFLD.

**Table 1.** Summary of studies describing the role of aryl hydrocarbon receptor in promoting and/or attenuating non-alcoholic fatty liver disease.

| | Model | Agonist/Antagonist | Genes | Effect | References |
|---|---|---|---|---|---|
| Constitutive AhR Activation | Constitutively active mouse-AhR mice | | ↑Cd36, Fatp1, Fatp2<br>↓Pparα, Acox1 | Disease promoting<br>↑Hepatic steatosis, liver triglycerides<br>↓Mitochondrial β-oxidation<br>↓Body mass, white adipose tissue fat | [15] |
| | Constitutively active human-AhR mice | | ↓Srebp1c, Acc1, Scd1, Fasn<br>↓Pparα, Cpt1α, Lcad, Mcad | Disease promoting<br>↑Hepatic steatosis, liver triglycerides & cholesterol<br>↓Body mass, white adipose tissue fat, mitochondrial β-oxidation | [81] |
| AhR Knockout | AhR KO mice | | | Disease promoting<br>↑Hepatic steatosis & fibrosis<br>↓Liver size | [53–55] |
| | AhR KO mice | | ↓Cd36, Srebp1c, Acc, Fasn<br>↓Tnfα, Il1β, Cd68 | Disease attenuating<br>↓Hepatic steatosis & fibrosis, body mass, white adipose tissue fat<br>↓Inflammation | [82,83] |
| | Inducible AhR conditional KO (AhR-iCKO) mice | | ↑Fgf21 | Disease attenuating<br>↓Hepatic steatosis, body mass | [84] |
| | Liver-specific AhR KO (AhR-LKO) mice | | ↑Srebp1c, Scd1, Acc1, Fasn, Gpam | Disease promoting<br>↑Hepatic steatosis, liver triglycerides, serum AST & ALT | [16] |
| Ligand-mediated AhR activation (Agonist) | C57BL/6 mice | 2,3,7,8-tetrachlorodibenzo-p-dioxin (TCDD) | ↑Acta2, Col1a1, Col1a2, α-Sma<br>↑Tgfβ, Fsp1, Tnfα, Il1β | Disease promoting<br>↑Liver fibrosis, serum ALT<br>↑Inflammation | [86] |
| | C57BL/6 mice, HepG2 cell lines | benzo[a]pyrene | ↑Cyp1a1, Srebp1c<br>↓Pparα | Disease promoting<br>↑Hepatic steatosis, liver triglycerides & cholesterol<br>↑Serum AST, ALT, triglycerides, cholesterol | [88] |

**Table 1.** *Cont.*

| | Model | Agonist/Antagonist | Genes | Effect | References |
|---|---|---|---|---|---|
| | C57BL/6 mice | Kynurenine | ↑Scd1, Tnfα, Il1β | Disease promoting<br>↑Liver fibrosis, serum ALT & AST | [89] |
| | C57BL/6 mice | 2-(1'H-indole-3'-carbonyl)-thiazole-4-carboxylic acid methyl ester (ITE) | ↓Acta2, Col1a1, Col1a2 | Disease attenuating<br>↓Liver fibrosis, serum ALT & AST | [76] |
| | AML12, HepG2 cell lines | Tryptamine<br>Indole-3-acetate | ↓Fasn, Srebp1c | Disease attenuating<br>↓Lipogenesis | [93] |
| | C57BL/6 mice | Indole-3-acetic acid | ↓Srebp1c, Scd1, Pparγ, Acaca, Gpam | Disease attenuating<br>↓Hepatic steatosis, liver triglycerides & cholesterol<br>↓ROS, SOD, MDA, GSH | [94] |
| | C57BL/6 mice, HepG2 cell line | Sulforaphane | ↓Srebp1c, Scd1, Acc1, Fasn<br>↓Tnfα, Mcp-1 | Disease attenuating<br>↓Hepatic steatosis, body mass, liver wt.<br>↓Serum AST, ALT, triglycerides, cholesterol | [95] |
| | C57BL/6 mice | Indole | ↓Acc, Fasn, Cpt1a<br>↓Tnfα, Il1β, Il-6 | Disease attenuating<br>↓Hepatic steatosis, plasma ALT | [96] |
| | AML12, HepG2 cell lines, C57BL/6 mice | Cinnabarinic acid | ↓CD36, Fasn, Srebp1, Scd1, Pparγ<br>↓Gpam, Gpat2, Dgat1, Dgat2, Mogat1, Tnfα, Tgfβ | Disease attenuating<br>↓Hepatic steatosis, liver triglycerides & cholesterol, serum ALT<br>↓Body mass ↓Inflammation | [97] |
| | C57BL/6 mice | 3,3'- diindolylmethane (DIM) | ↑Foxp3 | Disease attenuating<br>↓Hepatic steatosis, liver triglycerides & cholesterol, serum ALT<br>↓Body mass & Inflammation | [102] |
| Ligand-mediated AhR inactivation (Antagonist) | C57BL/6 mice, HepG2 cell line | Alpha-naphthoflavone | ↓Cyp1a1, Tnfα<br>↓Cyp1b1, Scd1, Spp1 | Disease attenuating<br>↓Hepatic steatosis<br>↓Serum AST, ALT, triglycerides & cholesterol<br>↓ROS, MDA<br>↑SOD, CAT, GSH | [90,91] |

**Table 1.** *Cont.*

| Model | Agonist/Antagonist | Genes | Effect | References |
|---|---|---|---|---|
| C57BL/6 mice | CH223191 | | Disease attenuating<br>↓Hepatic steatosis, body mass, white adipose tissue fat, serum triglycerides | [92] |

Upregulation and downregulation of genes and an increase and decrease in effect is indicated by an upward arrow (↑) and a downward arrow (↓), respectively. Genes: Acetyl-CoA carboxylase 1 (Acaca), acetyl-CoA carboxylase 1 (acc1), actin Alpha 2, smooth muscle (Acta2), acyl-CoA oxidase1 (Acox1), alpha-smooth muscle actin (α-Sma), carnitine palmitoyl transferase 1a (Cpt1a), carnitine palmitoyl transferase 2 (Cpt2), cluster of differentiation 36 (CD36), cluster of differentiation 68 (CD68), collagen type I alpha 1 chain (Col1a1), collagen type I alpha 2 chain (Col1a2), cytochrome p450 1a1 (Cyp1a1), cytochrome p450 1b1 (Cyp1b1), diacyl glycerol acyl transferase 1 (Dgat1), diacyl glycerol acyl transferase 2 (Dgat2), fatty acid transport protein 1 (Fatp1), fatty acid transport protein 2 (Fatp2), fatty acid synthase (Fasn), fibroblast growth factor 21 (Fgf21), fibroblast-specific protein 1 (Fsp1),forkhead box protein 3 (Foxp3), glycerol-3-phosphate acyltransferase, mitochondrial (Gpam), glycerol-3-phosphate acyltransferase 2 (Gpat2), interleukin 1 beta (Il1β), interleukin-6 (Il-6), long chain acyl-CoA dehydrogenase (Lcad), medium-chain acyl-CoA dehydrogenase (Mcad), monoacylglycerol o-acyltransferase 1 (Mogat1), monocyte chemoattractant protein-1 (Mcp-1), peroxisome proliferator-activated receptor alpha (Ppar-α), peroxisome proliferator- activated receptor gamma (Ppar-γ), secreted phosphoprotein (Spp1), stearoyl-CoA desaturase 1 (Scd1), sterol regulatory element-binding protein 1c (Srebp1c), transforming growth factor beta (Tgfβ), tumor necrosis factor alpha (Tnfα). Alanine aminotransferase (ALT), aspartate aminotransferase (AST), catalase (CAT), glutathione (GSH), malondialdehyde (MDA), reactive oxygen species (ROS), superoxide dismutase (SOD).

Therefore, beginning with the selection of the ligand, conformational changes in AhR, binding of cofactors, post-translational modifications that alter the chromatin architecture, and cross-talk of AhR signaling with other pathways leads to the simultaneous regulation of multiple signaling pathways—which ultimately can contribute to the potential attenuation or progression of NAFLD.

## 8. Summary and Conclusions

AhR, originally discovered as a receptor involved in xenobiotic metabolism, has recently been studied for its involvement in hepato-toxicity and -protection. Initial observations using AhR KO mice propelled our understanding of AhR's function in physiology and its pathological implications. Use of constitutively activated AhR models and exogenous agonists provided evidence towards exacerbated steatosis, fibrosis and other hallmarks of NAFLD. Corroborating the aforementioned observations, the tamoxifen-inducible knockdown of AhR and use of AhR antagonists alleviated NAFLD. However, recent studies showed that the activation of AhR signaling with selective endogenous AhR ligands, particularly indole derivatives from gut microbiome, and novel endogenous ligands such as cinnabarinic acid, can protect against fatty liver disease and possibly obesity. The role of AhR in NAFLD is therefore intricate and targeting AhR or its signaling pathway components for future drug development must take into consideration the characteristics of the ligand including binding affinity and duration of interaction with AhR, tissue/cell-specific activity of ligand, coactivators and corepressors interacting with AhR in response to ligand binding, epigenetic modifications, modulation of chromatin structure and cross-talk of AhR with other signaling pathways. It is also plausible that hitherto unknown factors and biochemical interactions may also regulate AhR's function in NAFLD. Finally, it is evident that AhR's involvement in fatty liver disease is complex as well as multifactorial, and a comprehensive biochemical and pathophysiological characterization is warranted

**Author Contributions:** Conceptualization, A.D.J.; writing—original draft preparation, N.Y.P.; writing—reviewing and editing, N.Y.P., J.E.F. and A.D.J.; funding acquisition, J.E.F. and A.D.J. All authors have read and agreed to the published version of the manuscript.

**Funding:** This research was funded by the National Institutes of Health-National Institutes of Diabetes and Digestive and Kidney Diseases, grant number R01 DK122028 to A.J. and R01 DK121951 to J.F.

**Institutional Review Board Statement:** Not applicable.

**Informed Consent Statement:** Not applicable.

**Data Availability Statement:** Not applicable.

**Conflicts of Interest:** The authors declare no conflict of interest.

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
