# Peer review of "Role of Hepatic Aryl Hydrocarbon Receptor in Non-Alcoholic Fatty Liver Disease"

_2813-2564, doi:10.3390/receptors2010001_

Round 1
Reviewer 1 Report
This review is an excellent overview of the state of the field with regards to AhR and NAFLD.
Table 1 gives a nice overview of the “double-edged sword” effects of AhR in NAFLD. I wonder if a model diagram could be developed to help show the conundrum pictorially.
The last sentence of the abstract and the last paragraph of the paper begin to talk about AhR as a drug target. Given the contradictory effects, it would be interesting to speculate, in this review, how that might occur. How might AhR, or its signaling pathway, be targeted for drug development in the case of NAFLD?
Author Response
We will like to thank the reviewer for evaluating our submitted article for publication in 'Receptors' journal. We greatly value the constructive feedback and suggestions provided by the reviewer which have helped us improve the quality of the article.
- Table 1 gives a nice overview of the “double-edged sword” effects of AhR in NAFLD. I wonder if a model diagram could be developed to help show the conundrum pictorially.
Thank you for the suggestion. We have now included a model diagram to help show the conundrum pictorially.
- The last sentence of the abstract and the last paragraph of the paper begin to talk about AhR as a drug target. Given the contradictory effects, it would be interesting to speculate, in this review, how that might occur. How might AhR, or its signaling pathway, be targeted for drug development in the case of NAFLD?
We have included a paragraph addressing this in the Summary and Conclusion section. Knowing that AhR’s involvement in fatty liver disease is complex and multifactorial, targeting AhR or its signaling pathway for drug development against NAFLD must take into consideration the characteristics of the ligand, coactivators and corepressors interacting with AhR in response to ligand binding, tissue/cell-specific activity of the ligand, epigenetic modifications, modulation of chromatin structure and cross-talk of AhR with other signaling pathways.
Reviewer 2 Report
Manuscript ID receptors-2022483 Role of hepatic Aryl hydrocarbon Receptor in Non-alcoholic fatty liver disease Nikhil Y. Patil , Jacob E. Friedman , Aditya D Joshi The review on the role of the AHR in NAFLD was well-written, fairly comprehensive, and very readable. Itt will be a reference used frequently. No major problems. My only complaints were that the authors seeseemed to have an aversion in using the identifier "the". When describing work from a given laboratory, the authors did not use "the", made up e.g., work was carried out in Smith lab that led to . . .; rather than: work was carried out in the Smith lab that . . . I found it disconcerting. Also, there were some subject verb agreement discrepancies. The review needs a careful and diligent editing of again minr issues.Author Response
We will like to thank the reviewer for evaluating our submitted article for publication in the 'Receptors' journal. We greatly value the constructive feedback and suggestions provided by the reviewer which have helped us improve the quality of the article.
- My only complaints were that the authors seemed to have an aversion in using the identifier "the". When describing work from a given laboratory, the authors did not use "the", made up e.g., work was carried out in Smith lab that led to . . .; rather than: work was carried out in the Smith lab that . . . I found it disconcerting. Also, there were some subject verb agreement discrepancies. The review needs a careful and diligent editing of again minor issues.
We have now added the identifier “the” wherever appropriate and corrected the subject-verb agreement discrepancies throughout the article. We have also carefully edited the article to make it grammatically correct.